# Self-Knowledge Without a Self? Learning Calibrated and Model-Agnostic Correctness Predictors from Historical Patterns

## Abstract

Generating reliable, calibrated confidence estimates is critical for deploying LLMs in high-stakes or user-facing applications, and remains an open challenge. Prior research has often framed confidence as a problem of eliciting a model's "self-knowledge", i.e., the ability of an LLM to judge whether its own answers are correct; this approach implicitly assumes that there is some privileged information about the answer's correctness that is accessible to the model itself. However, our experiments reveal that this assumption does not hold. Whether trained or training-free, an LLM attempting to predict the correctness of its own outputs generally performs no better than an unrelated model attempting the same task. In other words, LLMs have negligible self-knowledge for the purposes of correctness prediction. Moreover, we hypothesize that a key factor in predicting model correctness, i.e., building a "Correctness Model" (CM), is exposure to a target model's *historical predictions*. We propose multiple methods to inject this *historical* correctness information, including training an LLM to predict the confidences of many *other* LLMs, i.e., creating a Generalized Correctness Model (GCM). We first show that GCMs can be trained on the correctness of *historical predictions* from *many LLMs* and learn patterns and strategies for correctness prediction applicable across datasets and models. We then use CMs as a lens to study the source of the generalization and correctness prediction ability, adjusting their training data and finding that answer phrasing is a strong predictor for correctness. Moreover, our results suggest that a CM's ability to leverage world knowledge about answers for correctness prediction is a key enabler for generalization. We further explore alternative methods of injecting history without training an LLM, finding that including history as in-context examples can help improve correctness prediction, and post-hoc calibration can provide composable reductions in calibration error. We evaluate GCMs based on Qwen3-8B across 5 model families and the MMLU and TriviaQA datasets, as well as on a downstream selective prediction task, finding that reliable LLM confidence estimation is a generalizable and model-agnostic skill learned by systematically encoding correctness history rather than a model-specific skill reliant on self-introspection.[1]

## 1 Introduction

Confidence information is critical to understanding whether we should trust a system's response to a given query. For Large Language Models (LLMs), confidences enable us to understand honesty in a model (Kadavath et al., 2022), identify hallucinations (Zhou et al., 2025), route to experts when unconfident (Hu et al., 2024), rejection sample (Chuang et al., 2025), and even be leveraged as an RL signal to improve the quality of a model's behavior (Li et al., 2025b). Confidence calibration is the idea that we should enforce a desirable quality for confidences: a calibrated model's confidence should correspond to the empirical rate at which the model's responses are correct, i.e., outputting 90% confidence on an answer should correspond to a 90% chance of the answer being correct; confidence calibration is one desirable quality of confidences, which should be analyzed alongside performance measures such as accuracy.

---

[1]We will release our code and models on publication.

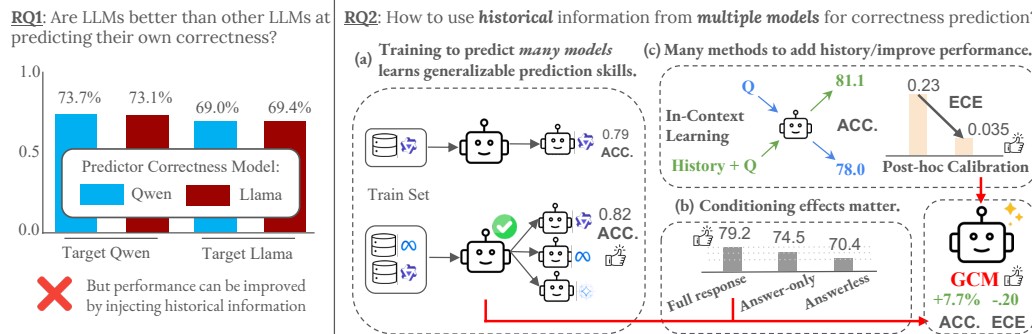

Figure 1: **RQ1 & RQ2 overview.** (**Left**) Self- vs. cross-model correctness prediction across Qwen and Llama: accuracies are comparable, suggesting no inherent advantage to a model *predicting its own* outputs. (**Right**) Historical information improves calibration: (a) training on multiple model's histories learns generalizable strategies for correctness prediction; (b) predictive power comes from phrasing of output, CM's world knowledge, and matching performance to question type; (c) History injected with post-hoc calibration and in-context learning helps improve correctness.

Many current approaches to LLM confidence estimation involve asking models to predict the correctness of their own responses, and are rooted in extracting the knowledge that LLMs have about their own correctness (Kadavath et al., 2022; Azaria & Mitchell, 2023; Li et al., 2024; Yin et al., 2023). To measure and improve the calibration of confidence estimates, these approaches also generally inherit frameworks and metrics from forecasting, where it is standard practice to calibrate forecasts of future events (Degroot & Fienberg, 1983; Guo et al., 2017a; Tian et al., 2023). However, a key component is missing in this forecasting analogy: *history*. Human forecasters attempt to calibrate themselves by explicitly recording their confidence on predictions over time and tracking systematic biases, which allows them to adjust and improve their performance (Mellers et al., 2015), albeit imperfectly. Unlike humans – who have privileged information about their own mental states and a memory of their past actions – current LLMs generally approach tasks without a running history mechanism for tracking historical performance. Moreover, when framing confidence estimation as a correctness prediction task, it is not clear that any given LLM is better-suited to predict its own correctness. In both cases, given a query $q$, a predicted response $r$ containing a predicted answer $\hat{r}$, the model is simply producing $P_\theta(\text{is\_correct}(\hat{r}) \mid q, r, \hat{r})$ there is no theoretical reason why this prediction should be better when the same LLM parameters $\theta$ were used to produce $r \sim P_\theta(r|q)$. In other words, it remains an open question as to whether models have *self-knowledge*.

We put these assumptions to the test by addressing two core research questions as outlined in Fig. 1. First, we ask **RQ1: Are LLMs better than other LLMs at predicting their own correctness?** Our experiments show that for the purposes of obtaining a calibrated confidence score (i.e. a calibrated $P(\text{is\_correct})$), **models have little to no privileged information about their own correctness**. For example, training Llama3.1-8B to predict its own confidence in being able to answer an MMLU question correctly results in the same performance as training Qwen2.5-7B to do the same, 69.35% vs 69.0% respectively (Fig. 1). We observe similar patterns in a training-free setting as well as when providing the answer and question together, indicating that using a model to predict its own confidences offers little to no performance advantage. This allows for the possibility of using one LLM to model the correctness of many others: removing reliance on self-knowledge, we can improve correctness prediction by learning from the history of many models. Indeed, in Fig. 1, we see that which model's history we train on is the clearest predictor for accuracy. Building on these findings, we ask **RQ2: What is the role of historical information from multiple models in calibrated correctness prediction**?

We explore these questions – and subsequent questions that follow from them – by constructing correctness models (CMs), i.e., models designed to provide calibrated $P(\text{is\_correct}(\hat{r})|\cdot)$ scores (which we also refer to as $P(c|\cdot)$ as a shorthand) predicting the correctness of target models (TMs). Unlike prior work, which has generally restricted CMs to the LLM generating responses – either in a zero-shot fashion (Tian et al., 2023) or via finetuning (Kapoor et al., 2024) – or used small linear classifiers (Liu et al., 2024; Kadavath et al., 2022), **we train LLMs on historical correctness**

**data from multiple different LLMs.** By varying the training data distribution, test settings, post-processing, and input features of the CM, we can concretely test questions and hypotheses about correctness estimation by examining the characteristics of the resulting CM. By building a variety of CMs, we investigate RQ1 and the following three axes of RQ2:

1. (RQ2A) **Generalization of CMs trained to predict multiple LLMs**: Do CMs trained on many models' outputs, referred to as Generalized Correctness Models (GCMs), learn generalized strategies for correctness prediction that transfer to other models and datasets? We find that CMs generalize well across different models families and model sizes, even outperforming self-emitted confidences of much larger OOD models, but less well across datasets (Section 3.2).

2. (RQ2B) **Conditioning factors relevant to prediction and generalization**: How do different conditioning variables (e.g., the question $q$, the response $r$, the predicted answer $\hat{r}$, or the target model's identity) affect correctness prediction and generalization ability? We measure the incremental gains from adding each variable and find that all components contribute meaningfully except the identity of the target model; interestingly, answer phrasing plays a substantial role. Moreover, improvements generalize across models, with the strongest generalization coming from parametric world-knowledge (Section 3.3).

3. (RQ2C) **Alternative methods of encoding history:** Does history incorporated in other ways help improve correctness? We study (a) *post-hoc calibration* and (b) *in-context learning*, which forgoes training in favor of supplying relevant prior examples in-context. We find injecting history via ICL examples helps improve correctness for larger models, and that using posthoc calibration to map historical confidences to correctness can help adapt a CM to dataset-wise OOD settings with few examples (Section 3.4).

Our research questions lead to practical insights about developing CMs: RQ2A shows that training Qwen3-8B on the aggregated correctness data from 8 models yields a GCM that outperforms the strongest single-model baseline (directly finetuning eight CMs, one on each target model) by 2.22% accuracy and .041 AUROC on average, observing an improvement on all target models for all metrics. Moreover, we show that the GCM based on Qwen3-8B outperforms the more powerful Llama3-70B's self-emitted confidences on MMLU by 2.4% absolute accuracy and .265 AUROC. Our GCM also outperforms Qwen3-32B's logit confidences, reducing ECE from .073 to .029 *without having been trained on Qwen3-32B or any other reasoning models*. The GCM transfers across datasets, outperforming a correctness model trained on the target dataset (a specific CM, or SCM) in terms of AUROC, and matches the SCM's ECE and accuracy after post-hoc calibration with as little as 5% of the SCM's training dataset. Finally, when applied to a downstream task such as selective prediction, we outperform Llama-3-70B's logit confidences and a SCM, enabling 30.0%, and 10.8% more coverage at a low 5% risk threshold respectively (See Section 3.5).

## 2 METHODS AND EXPERIMENTAL SETUP

**Correctness Models.** We define a Correctness Model as any system which can provide a confidence that a given query and response pair is correct. This allows us to treat methods such as prompting, probing, auxiliary models, finetuning, and posthoc calibrators all as parts of correctness models. Mathematically, a Correctness Model is any system that estimates the probability an answer is correct given a query $q$ and a response $r$ containing the answer $\hat{r}$, written as $P(\text{is\_correct}(\hat{r})|q, r, \hat{r})$. For LLMs, the query $q$ is the prompt and the response $r$ is the model's generation given the prompt. For MMLU, we make the distinction that $r$ refers to the model's entire response (average 198 tokens) and $\hat{y}$ refers only to the answer choice selected (A,B,C,D).

**Datasets.** Our main analysis is based on the MMLU dataset (Hendrycks et al., 2021) with additional dataset transfer experiments on the TriviaQA dataset (Joshi et al., 2017). To simulate a more realistic setting, we allow models to generate free form responses and use a judge model with ground truth access to grade them for correctness. We observe that across 8 models in the MMLU dataset, the **average response length was 198 tokens**, around one paragraph, with responses to math questions often containing reasoning traces that exceed 1000 tokens. A prompt, model response, and binary correctness label of whether the response was correct constitutes a correctness dataset, which is used in this work to inject historical correctness information into CMs. We build 18 correctness datasets by collecting responses from 10 separate models on the TriviaQA and MMLU datasets (8 from MMLU + 8 from TriviaQA + 2 models on MMLU for OOD testing). We include models

from the Gemma-3 (Team, 2025a), Qwen2.5 (Qwen et al., 2025), Qwen3 (Team, 2025b), Phi-3 (Microsoft, 2024) and Llama3 families (AIatMeta, 2024), as well as model sizes from 3B to 72B.

**Measuring Confidence.** Unless otherwise stated, we extract confidences from models via logit based confidences for all methods we study. We elicit **logit based confidences "P(True)"** (Kadavath et al., 2022) by measuring the probability of the token "yes" after exposure to a prompt, model name, and model response appended with the question "Please respond just 'yes' or 'no' in lowercase if the Response correctly answers the Prompt", when ablating the model response, this is rephrased to "Please respond just 'yes' or 'no' in lowercase if [Model Name] will respond correctly to Model Prompt:". Training examples in correctness datasets are structured according to this format with the ground truth yes/no appended. In Table 7 we ablate this prompt by removing the Model Name. Unless otherwise noted, **all models used in this work are instruction tuned models**.

**RQ1 Setup.** To address RQ1 (Section 3.1), we train two types of Correctness Models with different inputs. We train **Specific Correctness Models (SCMs)** by finetuning a LLM on a correctness dataset to predict the correctness of a response given a query. Excepting for when we explicitly tune these values during ablations, we use LoRA (Hu et al., 2021) with rank 32 and batch size 16 and train for 1 epoch on 70% of a correctness dataset, which is close to 10000 examples for datasets generated from both MMLU and TriviaQA. Unless otherwise noted, we initialize SCMs from a Qwen3-8B model. We utilize a specialized optimal batch size to obtain well calibrated ($\leq$ .03 ECE) Correctness Models out of the box with Cross Entropy Loss (see Section E). We train **Answerless Correctness Models** $P(c|q)$ (A finetuning based superset of P(IK) from Kadavath et al. (2022)) by finetuning a LLM to predict the probability that a target model will respond correctly to a query given only the query itself without the model response. Answerless CMs can also be seen as assessors (Hernández-Orallo et al., 2022). We use the same hyperparameters as the SCM.

**RQ2a Setup.** To analyze the generalization of correctness prediction strategies in RQ2a (Section 3.2) we introduce the General Correctness Model (GCM). We train **General Correctness Models (GCMs)** by finetuning a LLM, in this paper Qwen-3-8B, on the concatenation of 8 correctness datasets under the same training hyperparameters as the Specific Correctness Model. This trains the GCM to predict the correctness of many LLMs. We match the number of training datapoints and training steps between training one GCM to predict 8 LLMs vs training 8 SCMs to predict 8 LLMs, and further ablate impact of training steps in Table 24. Specifically, we train Qwen3-8B to predict Qwen2.5-3B to 72B, Llama3.1-8B, Qwen3-8B, Gemma-3-27B, and Llama-3-70B.

**RQ2b Setup.** To explore what parts of a correctness dataset contributes to correctness and what strategies generalize in RQ2b (Section 3.3), we ablate the GCM and SCM into Answerless Correctness Models, and further introduce an Answer-only model type on MMLU as an intermediate ablation. We train **Answer-only Correctness Models** $P(c|q, \hat{r})$ by extracting the answer choice letter from the target model's full response and training a SCM/GCM on the query and answer letter. See Table 1 for probabilistic representations. We also ablate model name information from a GCM (Section 2, Measuring Confidence).

**RQ2c Setup.** We further explore training free methods in RQ2c (Section 3.4) based on ICL verbalized confidences, and posthoc calibration. We inject **semantic ICL examples** into models by embedding the train split of a correctness dataset $(q, r, \hat{r}, c)$ into a vector database (see Section A for embedding details), and retrieving the top k=5 most semantically similar examples to the current example $(q, r, \hat{r})$ to inject into the prompt for the Correctness Model, we then elicit verbalized confidences. Since we do not focus on inference efficiency in this setting (5x-

Table 1: Settings compared in RQ2b, Section 3.3.

| Ablation name | Prob. Form |
| --- | --- |
| Full | $P(c|q, \hat{r}, r)$ |
| Answer-only | $P(c|q, \hat{r})$ |
| Answerless | $P(c|q)$ |

ing prompt length), we use verbalized confidences to give a further accuracy boost at the cost of efficiency. We elicit **verbalized confidences** (Tian et al., 2023) by prompting the model to give the "calibrated percent probability that the answer will be correct" in the format "xx.xx%". We **posthoc calibrate models** by holding out 5% of a correctness dataset and using the spline calibration (Lucena, 2018), beta-calibration (Kull et al., 2017), isotonic regression (Zadrozny & Elkan, 2002), or Platt scaling algorithms (Platt, 2000) to map raw model probabilities to calibrated probabilities.

**Evaluating Correctness Models.** We evaluate the performance of Correctness Models on 25% of any given correctness dataset, which is close to 3500 examples, ensuring the same questions are

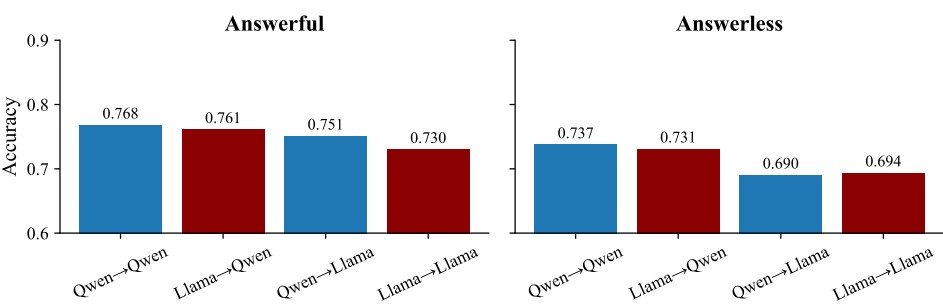

Figure 2: **Do LLMs possess special self-knowledge of their correctness?** We compare correctness prediction in *answerful* (with responses) and *answerless* (without responses) settings. Qwen2.5-7B beats Llama3.1-8B when responses are included, while both perform similarly without them, indicating that dataset signals and world knowledge drive performance, not privileged self-knowledge.

used across datasets to prevent train test contamination for GCMs. We highlight a CM's accuracy in predicting correctness as well as their expected calibration error, the standard metrics used for accessing the quality of predicted confidences (Guo et al., 2017b). Additionally, due to the variability of metrics like ECE (Guo et al., 2017b), we include the Root Mean Squared Calibration Error (Hendrycks et al., 2019) an adaptively binned measurement of calibration. We also include the Area Under the Curve of the Receiver Operating Characteristic (AUROC) which gives a more holistic estimate of predictive power. Importantly, this metric remains sensitive when data is class imbalanced, for example, when a large model such as Gemma-3-27b is correct on 78.8% of MMLU questions.

We include a comprehensive set of all prompts used for our analysis in Section D.

## 3 RESULTS

### 3.1 RQ1: MODELS HAVE NEGLIGIBLE SELF-KNOWLEDGE FOR CORRECTNESS ESTIMATION

The motivation for our work comes from the hypothesis that LLMs lack special information about their own correctness. We demonstrate this claim through several experimental settings, highlighting the two most illustrative settings. Given MMLU questions and responses, we first finetune both Qwen2.5-7B and Llama3.1-8B models to predict each other's correctness as well their own, with the results summarized in the Answerful setting of Fig. 2. We find that Qwen2.5-7B consistently predicts Llama3.1-8B's as well as its own correctness much better than Llama3.1-8B does. We attribute this to Qwen2.5-7B being a stronger model with greater parametric knowledge of the true answer to the MMLU questions (Qwen achieves 72% average MMLU accuracy whereas Llama3.1 only achieves 66%). This shows that using a stronger model is more critical to correctness prediction than "self-knowledge" stemming from using the same model for generation and verification.

To remove the effect of parametric knowledge, we repeat the experiment but remove the model response (answerless setting in Fig. 2), so that a greater parametric knowledge will not benefit Qwen2.5-7B. In this case we find that Qwen and Llama are roughly equally good at predicting Qwen's correctness, and the same is true when predicting Llama's correctness. If private knowledge existed (such as an internal confidence vector uniquely known to the model itself) we would expect that Llama would be able to predict its own confidences better. We further reinforce these findings by examining training-free settings and other pairs of models in Section C, where we find stronger models to be better predictors for correctness.

### 3.2 RQ2A: GENERALIZATION OF CMS TRAINED TO PREDICT MULTIPLE LLMS

Given that the self-knowledge of the LLM does not provide a significant advantage for a Correctness Model, we explore combining historical information from multiple models to improve CMs.

**Cross-Model Generalization.** We test whether correctness prediction learned from one model can transfer to others. A Qwen3-8B Generalized Correctness Model (GCM) trained as in Section 2 is evaluated on Llama3.1-8B and Gemma-3-27B against Specific Correctness Models (SCMs) trained

Table 2: Comparing Performance of different CMs on MMLU for predicting the correctness of Gemma3-27B and Llama3.1-8B. The General Correctness Model (GCM) outperforms all other baselines in terms of Accuracy and AUROC and achieves extremely low ECE $\leq$ .02.

| Method | Llama3.1-8B | | | | Gemma3-27B | | | |
|---|---|---|---|---|---|---|---|---|
| | Acc | ECE | RMSCE | AUROC | Acc | ECE | RMSCE | AUROC |
| P(True) | .741 | .219 | .253 | .807 | .789 | .197 | .301 | .707 |
| Verbal Confidence | .764 | .160 | .281 | .805 | .797 | .160 | .289 | .738 |
| ICL Verb. Conf. | .743 | .166 | .303 | .785 | .798 | .155 | .302 | .726 |
| Verb. Conf. (Qwen3-32B) | .780 | .161 | .244 | .833 | .807 | .166 | .272 | .725 |
| ICL Verb. Conf. (Qwen3-32B) | .811 | .103 | .186 | .862 | .833 | .119 | .194 | .796 |
| SCM (Trained On Target) | .792 | **.017** | **.069** | .857 | .796 | .037 | .091 | .811 |
| GCM | **.820** | .023 | .080 | .890 | .836 | .029 | .085 | .865 |
| GCM + Posthoc | .818 | .020 | .078 | **.890** | **.836** | **.016** | **.076** | **.865** |

Table 3: Comparing Performance of different CMs on TriviaQA for predicting the correctness of Gemma-3-27B and Llama3.1-8B. The General Correctness Model (GCM) outperforms all other baselines in terms of Accuracy by 1-4% and achieves extremely low ECE $\leq$ .023.

| Method | Llama3.1-8B | | | | Gemma3-27B | | | |
|---|---|---|---|---|---|---|---|---|
| | Acc | ECE | RMSCE | AUROC | Acc | ECE | RMSCE | AUROC |
| P(True) | .827 | .155 | .277 | .839 | .827 | .164 | .331 | .687 |
| Verbal Confidence | .834 | .136 | .323 | .821 | .825 | .158 | .344 | .687 |
| ICL Verb. Conf. | .827 | .119 | .234 | .855 | .826 | .145 | .254 | .755 |
| Verb. Conf. (Qwen3-32B) | .815 | .151 | .231 | .856 | .831 | .154 | .254 | .747 |
| ICL Verb. Conf. (Qwen3-32B) | .840 | .109 | .202 | .877 | .843 | .128 | .229 | .785 |
| Specific Model | .844 | .023 | .086 | .895 | .839 | .028 | .079 | .843 |
| General Model | .847 | .029 | .090 | .905 | .862 | .028 | .074 | .881 |
| General Model + Posthoc | **.8468** | .023 | **.077** | **.905** | **.862** | **.018** | **.072** | **.881** |

directly on each. With equal data and training time, the GCM outperforms SCMs by $\geq$3% accuracy on both and achieves $\leq$.03 ECE without post-hoc calibration (Table 2). We observe similar patterns for TriviaQA in Table 3. In Table 5, we confirm the GCM also outperforms Qwen3-8B trained to predict itself and in Table 4 show the same GCM outperforms Llama-3-70B's $P$(True) across all metrics. We next test on models *held out from training*. On Phi-3-mini, the GCM outperforms the SCM by 1.3% accuracy, .009 ECE, and .023 AUROC,[2] while on Qwen3-32B (also held out) it matches the SCM and surpasses Qwen3-32B's zero-shot $P$(True) (Table 6).[3] These results indicate that correctness prediction generalizes across families, sizes, and even held-out stronger models.

**Cross-Dataset Generalization.** Finally, we test the ability of the generalized model trained on MMLU to predict the correctness of models on TriviaQA in Table 8. We find that although the GCM achieves a similar AUROC to a SCM tuned on TriviaQA and outperforms P(True), it has a lower accuracy and a much higher ECE of .105 compared to the SCM's .023. Surprisingly, this suggests that capabilities generalize better across model families compared to datasets. We study generalizing similarities between models further in Section 3.3. Given the strength of the GCM in outperforming both SCMs and larger models in predicting the correctness of a variety of target models across datasets, we dedicate a section Section 3.5 to further evaluations of the General Correctness Model and its practical applications.

---

[2] Without training on any Phi-family models.

[3] Despite never being trained on reasoning-enabled models.

Table 4: Up Generalization: Qwen3-8B GCM vs. P(True) of Large ID Model (Llama-3-70B).

| | ID Large Model (Llama-3-70B) | | | |
|---|---|---|---|---|
| Method | Acc | ECE | RMSCE | AUROC |
| P(True) | .798 | .200 | .426 | .584 |
| GCM | .822 | .025 | .078 | .849 |

Table 5: Self Generalization: Qwen3-8B GCM vs. Qwen3-8B trained to predict itself.

| | Self Predict Model (Qwen3-8B) | | | |
|---|---|---|---|---|
| Method | Acc | ECE | RMSCE | AUROC |
| SCM | .814 | .035 | .091 | .835 |
| GCM | .834 | .021 | .071 | .867 |

Table 6: Out-of-Distribution Generalization. Qwen3-8B GCM predicting correctness on Phi-3-mini and Qwen3-32B, models that are held out from the GCM training set.

| | Phi-3-mini | | | | Qwen3-32B | | | |
|---|---|---|---|---|---|---|---|---|
| **Method** | Acc | ECE | RMSCE | AUROC | Acc | ECE | RMSCE | AUROC |
| P(True) (of target model) | .682 | .042 | .113 | .643 | .870 | .074, | .130 | .861 |
| Specific Model (trained on target) | .787 | .026 | .086 | .853 | **.873** | **.022** | **.072** | .876 |
| General Model (no exposure) | **.800** | **.017** | **.076** | **.876** | .871 | .029 | .084 | **.877** |

### 3.3 RQ2B: CONDITIONING FACTORS RELEVANT TO PREDICTION AND GENERALIZATION

**Ablating Conditional Distributions Used to Train Correctness Models.** We successively ablate the query $q$, the answer $\hat{r}$ and the full response $r$ from the correctness dataset to discover impact of each for both the SCM and GCM (Fig. 3). We interpret of each ablation as follows: The accuracy gap between $P(c|q,\hat{r},r)$ (Full) and $P(c|q,\hat{r})$ (Answer-only) ablates the *answer phrasing* of the target model's response, without removing its answer, showing the impact of learning correlations between how the answers are phrased and elaborated with accuracy. This ablation captures, for instance, the difference between seeing "I believe the answer is $4$", and just "$4$"; these findings align with work like Zhou et al. (2024), who study the importance of epistemic markers in confidence, and Stengel-Eskin et al. (2024b), who train LLMs to calibrate their use of linguistic signals that communicate confidence. The gap between $P(c|q,\hat{r})$ (Answer-only) and $P(c|q)$ (Answerless) ablates the target model's entire response, but preserves the query, showing the accuracy gain from allowing the CM to leverage its *world knowledge* to evaluate the likelihood that the answer $\hat{r}$ is correct independent of the past performance of the model on similar questions. Finally, the gap between $P(c|q)$ (Answerless) and $P(c)$ ablates the query, with $P(c|q)$ showing the performance gained by conditioning the target model's past performance on features of the questions compared to a model that simply predicts the majority class; this captures the notion that a given model may differ in its ability to answer different types of questions (Chen et al., 2025). We see a substantial increase in accuracy from every ablation, concluding that every ablated component, including response phrasing, is important to correctness prediction. By additionally comparing the SCM and the GCM, we find the GCM outperforms SCM by 2% accuracy in the answer-less setting, suggesting that there is some correlation between what questions LLMs most often answer correctly. The GCM improved 7% versus the SCM's 4% from answerless to Answer-only, showing that world-knowledge strategies for correctness prediction transfer well (Fig. 3).

**Role of Model Identity.** To test how much information about what model generated the response improves our ability to predict the correctness of target models, we remove the name of the target model from the prompt to the Answer-only GCM at training time, we find that while calibration and accuracy are impacted, it still outperforms the Answer-only SCM (Table 7). This suggests that much of the learned capability is model agnostic and not reliant on the identity of the target model.

### 3.4 RQ2C: ALTERNATIVE METHODS FOR ENCODING HISTORY

We observe in Section 3.1 that stronger models with more parametric knowledge can be better predictors of confidence. Moreover, we note that training the LLM is not always possible, especially with larger LLMs. This motivates us to consider injecting historical information in other ways. We explore two alternative methods: in-context learning (ICL) and post-hoc calibration.

Table 7: We ablate information about the identity of the target model from GCM, and discuss in Section 3.3.

| Method | Acc | ECE | RMSCE | AUROC |
|---|---|---|---|---|
| GCM Answer-only | .789 | .034 | .088 | .852 |
| −Name Ablated | .763 | .034 | .091 | .847 |
| SCM Answer-only | .745 | .023 | .087 | .810 |

Table 8: Out-of-Distribution Generalization. GCM trained on MMLU, tested on TriviaQA.

| Method | Acc | ECE | RMSCE | AUROC |
|---|---|---|---|---|
| P(True) | .827 | .155 | .277 | .839 |
| SCM (TriviaQA) | .844 | **.023** | **.080** | .895 |
| GCM (MMLU) | .828 | .105 | .150 | .896 |
| GCM + Posthoc | **.844** | .031 | .088 | **.896** |

**In-Context Learning.** Rather than training a CM on a dataset of historical examples, we embedding the training split of the target model's correctness dataset $(q, r, \hat{r}, c)$ and the current example $(q, r, \hat{r})$, retrieving top k=5 similar training examples to include in-context (details in Section 2). We show in Table 2 that injecting semantically relevant examples from the correctness dataset via ICL improves accuracy by 4.6% and reduces ECE by 7.8% when predicting Gemma3-27B's performance with Qwen3-32B, compared to verbalized confidences without ICL. As the ICL setting focuses less on inference efficiency, multiplying prompt length by k, we allowed the model to verbally reason about correctness to further improve accuracy at the cost of inference time. However, Qwen3-8B showed no gains, suggesting a minimum base capability is needed for ICL benefits.

**Posthoc Calibration.** Posthost calibration injects historical information by directly aligning an CM's output confidences with the historical ground truth $P(c)$ without conditioning on $q, r$ or $\hat{r}$, as in Section 2. Recall that in RQ2a (Section 3.2), we showed that transfer to new datasets is harder for a GCM: although we outperformed the target SCM in terms of AUROC, the GCM had more than .10 ECE after transfer. However, we find calibrating the result increases accuracy and decreases ECE to match performance of the SCM (Table 8) using only 5% of the target dataset's samples. In Table 8, we show that it is possible to substantially reduce calibration error by 0.105 to 0.031 with 5% of the dataset. We additionally observe that it is possible to further calibrate the GCM's output probabilities to reach even lower ECE with posthoc calibration (Table 2).

### 3.5 RECOMMENDATIONS FOR PERFORMANT CORRECTNESS PREDICTION

**Building the Most Performant Correctness Model.** Here, we put together the findings from Section 3.2, Section 3.3, and Section 3.4 to summarize the best practices for building a GCM. We recommend the GCM with posthoc calibration as an accurate and calibrated correctness prediction method. In Table 2 we found that the GCM significantly outperforms strong baselines in distribution, and transfers without training to beat models trained on OOD target models of different model families, as well as reasoning models Table 6. In addition, when combined with posthoc calibration, it beats SCMs trained on an OOD target dataset in terms of AUROC, matching it in terms of accuracy and ECE Table 8. Further, the GCM is a inference efficient prefill only method, requiring less than 0.125 seconds to process the correctness of an average MMLU response with 200 tokens and 7.3 minutes to process 3511 examples. This solidifies the GCM with Posthoc calibration as our recommend method of modeling correctness given history. If training is not possible, we recommend using the ICL method presented in Section 3.4. However, we note that while ICL on a significantly stronger model (Qwen3-32B) can match the predictive accuracy of a GCM based on Qwen3-8B, it suffers from high calibration error and has much lower AUROC, which is important for downstream applications such as re-ranking and selective prediction. Additionally, the inference cost of ICL is significantly higher in terms of latency, compute, and memory requirements, due to requiring a large base model, multiplying input prompt length by $k$ for $k$ retrievals, and requiring the generation of a reasoning chain. One MMLU evaluation run (3500 examples, ∼2.6s/example) already exceeds the cost of training a SCM on correctness.

**Downstream Evaluation on Selective Prediction.** Here, we show that the GCM also provides downstream benefits in a selective prediction task. Selective prediction requires a system to selectively abstain from examples that are unlikely to be correct, with the objective of maximizing coverage (the percentage of examples for which an answer is produced) while minimizing risk (the percentage of predicted answers that were incorrect). Intuitively, the trade-off between coverage and risk is one between usability and safety, with full coverage (no abstention) system having high usability but low safety, while abstaining on all examples (zero coverage) incurs no risk but represents

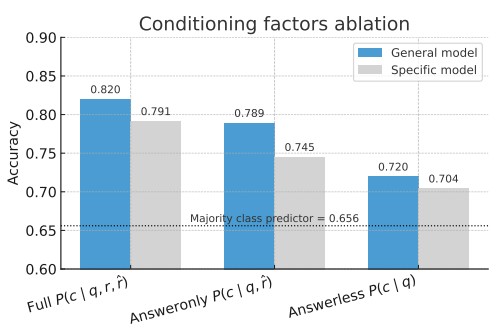

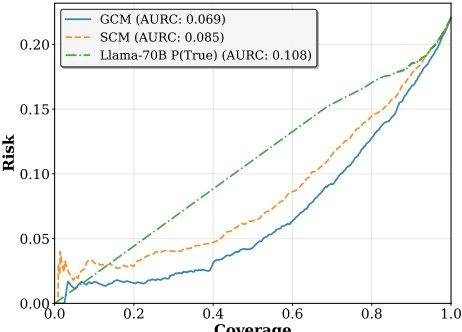

Figure 3: **Conditioning factors** ablation. GCMs and SCMs across conditioning settings in RQ2b (Section 3.3). More metrics: Table 25.

Figure 4: **Risk-Coverage Curves** for Selective Prediction, **lower** AURC curves are better.

a useless model. Fig. 4 shows the risk-coverage curves for the GCM, SCM, and for Llama3-70B ; here, a lower AURC indicates a better trade-off between coverage and risk. As shown in Fig. 4, our results indicate that compared to the target model's self emitted confidences or a model-specific SCM, the generalized model consistently achieves lower risk (y-axis) at the same level of coverage (x-axis Fig. 4). This suggests that the GCM produces more reliable predictions, making it better suited for robust deployment.

## 4 RELATED WORK.

**Self-Knowledge and Confidence Calibration.** Calibration, crucial for deciding when to trust AI systems, has been studied in neural models (Naeini et al., 2015; Guo et al., 2017a; Ovadia et al., 2019; Wang et al., 2020a) and more recently in LLMs (Mielke et al., 2022; Kadavath et al., 2022; Kuhn et al., 2023; Stengel-Eskin et al., 2024a; Tian et al., 2023). Early work showed models like T5, BART, and GPT-2 are poorly calibrated on QA, motivating post-hoc and fine-tuning methods (Jiang et al., 2021). Other studies examined overconfidence in dialogue (Mielke et al., 2022), prompting-based calibration (Kadavath et al., 2022), and fine-tuning (similar to SCMs) with correctness labels (Kapoor et al., 2024). Further efforts probed unanswerable questions (Yin et al., 2023), lying behavior via hidden activations (Azaria & Mitchell, 2023), and black-box elicitation through prompting, sampling, and aggregation (Xiong et al., 2024). Yet LLMs remain overconfident: calibration improves with scale but still lacks reliability. Some evidence exists that LLMs may have privileged access to some aspects of their behavior, such as preferences, and detection of activation steering (Binder et al., 2025; Lindsey, 2025). In contrast, we show models lack privileged access to their own correctness information and propose a more general solution to calibrate *multiple* LLMs at once.

**Correctness Models and Cross-Model Transfer.** Another line of work uses *correctness models (CMs)* to predict whether a response is correct. The simplest rely on self-reported confidence (Tian et al., 2023), while stronger methods probe hidden states (Liu et al., 2024; Kadavath et al., 2022; Beigi et al., 2024; Azaria & Mitchell, 2023) or fine-tune LLMs directly on correctness tasks (Kapoor et al., 2024). Recent efforts capture *semantic uncertainty*, modeling meaning variability for better correctness correlation (Kuhn et al., 2023). Surrogate approaches also show promise: Shrivastava et al. (2023) report that even untrained LLaMA models can outperform GPT's self-reported probabilities, revealing biases in elicitation. These studies suggest correctness signals can transfer across models, but focus on one-to-one transfer. In contrast, we identify the key factors shaping CM calibration and introduce a Generalized Correctness Model (GCM) that aggregates correctness patterns across many models for more robust prediction. See Section H for more details on related works.

**Reward Models, LLM Judges, and Quality Estimation.** While most LLM reward models (RMs) provide pairwise judgements to align with human preference, closest to our work, LLM Judges, Outcome Reward Models, and Process Reward Models are trained to predict the expected reward of full or partial reasoning outputs; when the reward is accuracy-based, these models produce rewards or Likert judgments reflecting answer correctness (Lambert et al., 2025; Tan et al., 2025; Luo et al., 2024; Cobbe et al., 2021; Zheng et al., 2025). Quality estimation (QE) is another line

of work have been applied to estimate the quality of a response without ground truth reference, the predominant application focuses on judging the quality of machine translation outputs (Specia et al., 2009; Zhao et al., 2024). By design, RMs/QE do not consider historical information as they are trained with no knowledge of which model produced the answers they are judging (Lambert et al., 2025; Tan et al., 2025; Zheng et al., 2025). This is necessary to produce unbiased RM/QE models: for example, outputs should not be rewarded just because they come from a model that typically produces correct outputs. In contrast, we focus on learning from "historical patterns", conditioning on the identity of the models generating a response, as well as training on an unaltered history of model performance. Concretely, this allows us to learn both tasks that a specific model is strong/weak in as well as the general difficulty of types of questions for LLMs. Consider any case where the CM/RM/QE has almost no parametric understanding of the true answer (e.g., extremely difficult math), the RM/QE performance would reduce to random as parametric understanding approaches zero (Tan et al., 2025), whereas the Answerless GCM in Fig. 3 shows the nontrivial performance of a CM (+22% above random). Additionally, the work on RM/QE generally does not consider calibrated confidences, RMs output unbounded scalars and QE's ground truth is generally quality, which lacks probabilistic meaning for calibration.

**Assessor Models.** Assessor models are trained to predict the correctness of a model's potential response given only the query (Hernández-Orallo et al., 2022). Assessor models are useful for the purposes of routing to the most capable model for a given query and preemptive rejection when it is clear that a model has a low chance of responding correctly (Pacchiardi et al., 2025; Zhou et al., 2022; Pacchiardi et al., 2024). Assessors can be seen as an ablation of our method that does not consider responses, which we refer to as Answerless Correctness Models (Section 2). Moreover, they are also similar to the P(IK) setting in the confidence calibration literature (Kadavath et al., 2022). We analyze Answerless Correctness Models / Assessors as part of a hierarchy of conditioning factors for correctness prediction in Section 3.3.

## 5 CONCLUSION

The insight that LLMs have no self-knowledge is counterintuitively beneficial for the purposes of predicting the correctness of LLMs. We find that a General Correctness Model based on a LLM, trained to predict the correctness of many LLMs, is able to generalize and learn transferable correctness prediction strategies across a variety of models, suffering no penalty for predicting models apart from itself. A GCM outperforms both models trained to predict their own correctness, and the self-emitted correctness confidences of larger models the GCM has not trained on.

## ETHICS STATEMENT

By addressing calibration – an important ingredient for developing safer AI systems – we believe our work will have a positive impact in improving ethical and safety considerations. We do not foresee any additional ethical implications beyond standard ethical and safety considerations that apply to AI research generally.

## REPRODUCIBILITY STATEMENT

We detail our experimental setup in Section 2 and provide an expanded version in Section A to guide the reproducibility of our experiments and methods introduced in this work.

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

## A ADDITIONAL DETAILS ABOUT EXPERIMENTAL SETUP

**ICL Retrieval Details.** In order to facilitate semantic retrieval for the **semantic ICL examples** setting (Section 2), we utilize the chroma library (Chroma, 2025), and we use the default embed function, which at time of writing is "all-MiniLM-L6-v2" based on Wang et al. (2020b).

## B DISCUSSION

### B.1 DISCUSSION ON CHATGPT'S MEMORY SYSTEM AND SIMILAR TECHNIQUES FOR INJECTING HISTORY

We discussed the lack of historical information for LLM based systems in Section 1. We would like to point out that systems such as ChatGPT incorporates a history function. However, we make the distinction that what is necessary is to inject *historical correctness information*, not simply historical information. Additionally, systems such as ChatGPT preserve sparse memories that do not always give a direct account of the performance of their own previous generations, or indeed, even the generations themselves.

## C FURTHER RESULTS SHOWING THAT MODELS HAVE NEGLIGIBLE SPECIAL INFORMATION ABOUT THEIR OWN ABILITIES

See Tables 9 - 22 for a consolidated comparison of accuracy, calibration (ECE/RMSCE) and AUROC across answerless, answerful, and untrained settings, covering both within-model and cross-model transfers. We conduct tests across 5 model families and model sizes ranging from 3B to 72B. The training hyperparameters used follow that of Specific Correctness Models (SCMs Section 2). We always conduct nxn tests in order to make relative differences clear and ensure that confounding variables like how lora interacts with a specific architecture/size are not obstacles to spotting privileged self-prediction capability. In our tests, we see no particular advantage to self prediction with any consistency.

Qwen2.5-72B and Llama3-70B have the same number of hidden layers and hidden size (this means the interaction of lora with these two models might be expected to be very similar in terms of

optimization and representation power), as such Table 15 - 18 may be of particular interest and show clearly that there is no particular advantage to self-prediction.

Table 9: Untrained setting (row-wise). No tuning or epistemic supervision used.

| Configuration | Acc | ECE | RMSCE | AUROC |
|---|---|---|---|---|
| Qwen2.5-7B→Qwen2.5-7B | 0.6380 | 0.2720 | 0.2213 | 0.5649 |
| Llama3.1-8B→Qwen2.5-7B | 0.7075 | 0.2041 | 0.1933 | 0.6556 |
| Qwen2.5-7B→Llama3.1-8B | 0.5523 | 0.3312 | 0.2421 | 0.5197 |
| Llama3.1-8B→Llama3.1-8B | 0.6568 | 0.2916 | 0.2289 | 0.6679 |

Table 10: Answerless setting (row-wise) with Qwen3-8B and Qwen2.5-7B.

| Configuration | Acc | ECE | RMSCE | AUROC |
|---|---|---|---|---|
| Qwen2.5-7B→Qwen2.5-7B | 0.7277 | 0.0181 | 0.0888 | 0.7194 |
| Qwen3-8B→Qwen2.5-7B | 0.7371 | 0.0287 | 0.0923 | 0.7513 |
| Qwen2.5-7B→Qwen3-8B | 0.7488 | 0.0225 | 0.0855 | 0.7138 |
| Qwen3-8B→Qwen3-8B | 0.7650 | 0.0364 | 0.0937 | 0.7561 |

Table 11: Answerless setting (row-wise), grouped by *target model*.

| Configuration | Acc | ECE | RMSCE | AUROC |
|---|---|---|---|---|
| Qwen2.5-7B→Qwen2.5-7B | 0.7374 | 0.0160 | 0.0810 | 0.7297 |
| Llama3.1-8B→Qwen2.5-7B | 0.7308 | 0.0235 | 0.0857 | 0.7372 |
| Qwen2.5-7B→Llama3.1-8B | 0.6901 | 0.0242 | 0.0908 | 0.7027 |
| Llama3.1-8B→Llama3.1-8B | 0.6935 | 0.0163 | 0.0837 | 0.7282 |

Table 12: Answerful setting (row-wise). Models are given access to the predicted answer.

| Configuration | Acc | ECE | RMSCE | AUROC |
|---|---|---|---|---|
| Qwen2.5-7B→Qwen2.5-7B | 0.7679 | 0.0189 | 0.0805 | 0.7910 |
| Llama3.1-8B→Qwen2.5-7B | 0.7610 | 0.0242 | 0.0787 | 0.7900 |
| Qwen2.5-7B→Llama3.1-8B | 0.7508 | 0.0219 | 0.0777 | 0.8039 |
| Llama3.1-8B→Llama3.1-8B | 0.7300 | 0.0205 | 0.0842 | 0.7749 |

Table 13: Answerless setting (row-wise) with Qwen2.5-72B and Meta-Llama-3-70B.

| Configuration | Acc | ECE | RMSCE | AUROC |
|---|---|---|---|---|
| Qwen2.5-72B→Qwen2.5-72B | .831 | .029 | .087 | .763 |
| Meta-Llama-3-70B→Qwen2.5-72B | .828 | .027 | .091 | .732 |
| Qwen2.5-72B→Meta-Llama-3-70B | .782 | .028 | .080 | .753 |
| Meta-Llama-3-70B→Meta-Llama-3-70B | .776 | .028 | .102 | .747 |

Table 14: Answerful setting (row-wise) with Qwen2.5-72B and Meta-Llama-3-70B.

| Configuration | Acc | ECE | RMSCE | AUROC |
|---|---|---|---|---|
| Qwen2.5-72B→Qwen2.5-72B | .846 | .025 | .087 | .838 |
| Meta-Llama-3-70B→Qwen2.5-72B | .844 | .025 | .086 | .784 |
| Qwen2.5-72B→Meta-Llama-3-70B | .844 | .029 | .087 | .875 |
| Meta-Llama-3-70B→Meta-Llama-3-70B | .807 | .029 | .100 | .788 |

| Target \ Correctness | Qwen-2.5-72B | Llama-3-70B | Gemma-3-27B |
|---|---|---|---|
| Qwen-2.5-72B | 0.831 | 0.828 | 0.827 |
| Llama-3-70B | 0.782 | 0.776 | 0.774 |
| Gemma-3-27B | 0.784 | 0.784 | 0.791 |

Table 15: Additional RQ1 Large Models Experiments (Answerless) — Accuracy (Acc)

| Target \ Correctness | Qwen-2.5-72B | Llama-3-70B | Gemma-3-27B |
|---|---|---|---|
| Qwen-2.5-72B | 0.029 | 0.027 | 0.031 |
| Llama-3-70B | 0.028 | 0.028 | 0.037 |
| Gemma-3-27B | 0.030 | 0.026 | 0.029 |

Table 16: Additional RQ1 Large Models Experiments (Answerless) — Expected Calibration Error (ECE)

## D  PROMPT TEMPLATES

We document the prompt templates generated under different argument settings in the following paragraphs.

**SCM / GCM / Answer-only / P(True)**

```
You are grading [model name]'s responses to prompts for correctness.
###Model Prompt
{input_prompt}
###Model Response
{cleaned_model_completion}
###Instruction
Please respond just 'yes' or 'no' in lowercase if Model Response
correctly answers Model Prompt: {ground truth yes/no}
```

(Note: the trailing "yes" or "no" is inserted according to ground truth "is_correct" of the response when used as a training example, when used during inference, it is removed.)

**Answerless Setting**

```
You are predicting the likelihood that [model name] will respond
correctly to a prompt.
###Model Prompt
{input_prompt}
###Instruction
Please respond just 'yes' or 'no' in lowercase if [model name] will
respond correctly to Model Prompt: {ground truth yes/no}
```

(Note: the trailing "yes" or "no" is inserted according to ground truth "is_correct" of the response when used as a training example, when used during inference, it is removed.)

**Verbalized Confidence**

```
You are examining the correctness of [model name]'s predictions.
You are given a question and answer pair and you must output the
probability that [model name] has answered the question correctly.
You should make a concise and informative analysis of the current
question and response to determine whether the response has
correctly answered the question.
After your analyses, please respond with a calibrated percent
probability that the answer will be correct in the format
ANSWER_CORRECT_PROBABILITY: xx.xx%
```

| Target \ Correctness | Qwen-2.5-72B | Llama-3-70B | Gemma-3-27B |
|---|---|---|---|
| Qwen-2.5-72B | 0.846 | 0.844 | 0.843 |
| Llama-3-70B | 0.844 | 0.807 | 0.822 |
| Gemma-3-27B | 0.849 | 0.817 | 0.825 |

Table 17: Additional RQ1 Large Models Experiments (Answerful) — Accuracy (Acc)

| Target \ Correctness | Qwen-2.5-72B | Llama-3-70B | Gemma-3-27B |
|---|---|---|---|
| Qwen-2.5-72B | 0.025 | 0.025 | 0.024 |
| Llama-3-70B | 0.029 | 0.029 | 0.016 |
| Gemma-3-27B | 0.023 | 0.033 | 0.022 |

Table 18: Additional RQ1 Large Models Experiments (Answerful) — Expected Calibration Error (ECE)

```
##Current Model Prompt
{input_prompt}
##Current Model Response
{cleaned_model_completion}
##Please respond with with a verbalized and calibrated percent
probability that the Current Model Response is a correct response
to Current Model Prompt and output your answer in the EXACT format
'ANSWER_CORRECT_PROBABILITY: xx.xx%'.
Be sure to follow the format exactly.
```

**ICL Verbalized Confidence**

```
You are examining the correctness of [model name]'s predictions.
You are given {N} training datapoints consisting of questions
[model name] has been asked in the past.
Training datapoints contain a question, [model name]'s response,
and human labeled yes/no of whether the response was correct.
After the training datapoints you are given the current question
and answer pair and you must output the probability that
[model name] has answered the question correctly.
You should make a concise and informative analysis of the current
question and response to determine whether the response has
correctly answered the question.
Then, if you are still unsure of your decision, you can explicitly
analyze the model's past performance on similar examples and make
appropriate adjustments depending on the relevance of the training
examples.
After your analyses, please respond with a calibrated percent
probability that the answer will be correct in the format
ANSWER_CORRECT_PROBABILITY: xx.xx%

##Previous Performances
Example 0 -- Distance: {d_0} (lower = more similar)
{document_0}

...
Example N -- Distance: {d_N} (lower = more similar)
{document_N}

##Current Model Prompt
{input_prompt}
##Current Model Response
```

| Target \ Correctness | Gemma-3-4B | Gemma-3-12B | Gemma-3-27B |
|---|---|---|---|
| Gemma-3-4B | 0.687 | 0.817 | 0.831 |
| Gemma-3-12B | 0.772 | 0.796 | 0.829 |
| Gemma-3-27B | 0.794 | 0.812 | 0.825 |

Table 19: Additional RQ1 Adjacent Gemma Size Experiments (Answerful) — Accuracy (Acc)

| Target \ Correctness | Gemma-3-4B | Gemma-3-12B | Gemma-3-27B |
|---|---|---|---|
| Gemma-3-4B | 0.028 | 0.031 | 0.017 |
| Gemma-3-12B | 0.026 | 0.028 | 0.022 |
| Gemma-3-27B | 0.025 | 0.025 | 0.022 |

Table 20: Additional RQ1 Adjacent Gemma Size Experiments (Answerful) — Expected Calibration Error (ECE)

```
{cleaned_model_completion}
##Please respond with with a verbalized and calibrated percent
probability that the Current Model Response is a correct response
to Current Model Prompt and output your answer in the EXACT format
'ANSWER_CORRECT_PROBABILITY: xx.xx%'.
Be sure to follow the format exactly.
```

(The ICL Verbalized Confidence prompt is not used for training, and thus does not include ground truth labels, except in the included ICL examples, during any form of its usage.)

**Model Name Ablation**

```
You are grading responses to prompts for correctness, responses
could be generated from multiple LLMs.
###Prompt
{input_prompt}
###Response
{cleaned_model_completion}
###Instruction
Please respond just 'yes' or 'no' in lowercase if the Response
correctly answers the Prompt: {ground truth yes/no}
```

(Note: the trailing "yes" or "no" is inserted according to ground truth `is_correct` of the response when used as a training example, when used during inference, it is removed.)

**Notes for Reproducibility**

- Variable placeholders ({input_prompt}, {cleaned_model_completion}, {document_i}, {d_i}, {ground truth yes/no}) are filled dynamically from the correctness dataset.

- {document_i} and {d_i} refer to results from ICL retrieval described under the ICL Retrieval Details heading, they represent a full training example text with labels formatted according to the SCM/GCM/Answer-only/P(True) format and the rank of its similarity to the current example under consideration respectively.

- {input_prompt} and {cleaned_model_completion} refer to the MMLU/TriviaQA prompt to the model, and the target model's response.

- For the Answer-only prompt in MMLU, the {cleaned_model_completion} is truncated to only display the answer choice letter.

| Target / Correctness | Mistral-7B | Qwen2.5-7B | Llama-3.1-8B | Qwen3-8B |
|---|---|---|---|---|
| Mistral-7B | 0.686 | 0.669 | 0.681 | 0.670 |
| Qwen2.5-7B | 0.731 | 0.728 | 0.726 | 0.738 |
| Llama-3.1-8B | 0.691 | 0.681 | 0.677 | 0.702 |
| Qwen3-8B | 0.751 | 0.749 | 0.750 | 0.766 |

Table 21: Additional RQ1 Experiments with varied models (Answerless) - Accuracy (Acc)

| Target / Correctness | Mistral-7B | Qwen2.5-7B | Llama-3.1-8B | Qwen3-8B |
|---|---|---|---|---|
| Mistral-7B | 0.024 | 0.025 | 0.018 | 0.026 |
| Qwen2.5-7B | 0.014 | 0.024 | 0.028 | 0.032 |
| Llama-3.1-8B | 0.021 | 0.022 | 0.022 | 0.027 |
| Qwen3-8B | 0.023 | 0.020 | 0.017 | 0.035 |

Table 22: Additional RQ1 Experiments with varied models (Answerless) - ECE

# E    OPTIMAL BATCH SIZE LEADS TO NEGLIGIBLE CALIBRATION ERROR

**Minimizing ECE with Training Batch Size.**    Another analysis we make is regarding the effect of the training batch size on calibration. Prior work have sometimes attributed miscalibration to the use of the Cross Entropy Loss or otherwise suggested that a different loss function should be used to ensure calibrated models after finetuning (Mukhoti et al., 2020; Damani et al., 2025; Li et al., 2025c). For our particular experimental setting (training SCMs and GCMs), we find that batch size has a surprising effect on calibration, and that by carefully setting the batch size we can overcome the miscalibration issue caused by CEL to reach a negligible .01-.02 ECE. We observe that a small batch size of 1 is especially detrimental and higher batch sizes than 32 can also harm ECE. We build our SCMs and GCMs using a batch size of 16 based on this observation (Table 23).

# F    UNLIMITED TRAINING TIME ABLATION

For our main analysis we train the General Model for the same number of epochs as the specific model to match training time. In such a case, training multiple specific models and training one general model would have approximately the same training time cost. We show an ablation here, that even given an unlimited amount of training time (until overfitting occurs), the GCM still outperforms SCMs (Table 24).

# G    CONDITIONING FACTORS ABLATIONS

We include the full metrics for conditioning factors ablations in appendix Table 25.

# H    RELATED WORK

**Self-Knowledge and Confidence calibration.**    Since calibration is essential for deciding when to trust AI systems, prior work has extensively studied calibration in neural models (Naeini et al., 2015; Guo et al., 2017a; Ovadia et al., 2019; Wang et al., 2020a), with more recent efforts turning to calibration in large language models (LLMs) (Mielke et al., 2022; Kadavath et al., 2022; Kuhn et al., 2023; Stengel-Eskin et al., 2024a; Tian et al., 2023). Early studies found that generative models such as T5, BART, and GPT-2 are often poorly calibrated for QA tasks, requiring post-hoc or fine-tuning methods to better align probabilities with correctness (Jiang et al., 2021). Other works examined overconfidence in dialogue agents and proposed linguistic calibration, matching expressions of doubt with correctness likelihoods, as a remedy (Mielke et al., 2022). Prompting-based methods have also been explored: Kadavath et al. (2022) showed that larger LLMs can produce reasonably calibrated probabilities when asked directly, while Kapoor et al. (2024) argued that prompting alone is insufficient, and that fine-tuning with correctness labels yields better transferable estimates. Additional studies examined unanswerable questions (Yin et al., 2023), lying behavior via hidden activations (Azaria & Mitchell, 2023), and black-box elicitation frameworks combining

Table 23: Uncalibrated accuracy and ECE by `gdacc` for both Alpha models.

| Model | gdacc | Uncal Acc | Uncal ECE |
|---|---|---|---|
| Qwen2.5-7B | 128 | .750 | .102 |
| Qwen2.5-7B | 64 | .780 | .039 |
| Qwen2.5-7B | 32 | .788 | .030 |
| Qwen2.5-7B | 16 | .792 | .025 |
| Qwen2.5-7B | 4 | .798 | .066 |
| Qwen2.5-7B | 2 | .803 | .118 |
| Qwen2.5-7B | 1 | .810 | .146 |

Table 24: **Unlimited training time ablation**. Columns report Accuracy (avg_correct), Binary ECE (↓), RMSCE (↓), and AUROC (↑).

| Method | Acc ↑ | ECE ↓ | RMSCE ↓ | AUROC ↑ |
|---|---|---|---|---|
| Optimal SCM | .8223 | .0232 | .0728 | .8936 |
| Optimal GCM | .8448 | .0348 | .0874 | .9122 |

prompting, sampling, and aggregation (Xiong et al., 2024). Despite these advances, LLMs remain overconfident, and calibration quality improves with scale but falls short of reliability. In contrast to these self-knowledge-based approaches, our work demonstrates that models lack privileged access to their own correctness and introduces a more general solution to calibrate *multiple* LLMs at once.

**Correctness Models and Cross-Model Transfer.**    A parallel line of work explicitly uses *correctness models (CMs)* to estimate whether a response is correct. The simplest CMs rely on self-reported confidence from the model itself (Tian et al., 2023), while stronger approaches train linear probes on hidden states (Liu et al., 2024; Kadavath et al., 2022; Beigi et al., 2024; Azaria & Mitchell, 2023) or fine-tune entire LLMs to answer correctness questions directly (Kapoor et al., 2024). Recent studies go beyond surface calibration by modeling *semantic uncertainty*, capturing variability in the meaning of generated outputs, which has been shown to better correlate with correctness (Kuhn et al., 2023). Another intriguing development is the use of surrogate models: Shrivastava et al. (2023) find that even untrained LLaMA models can sometimes predict GPT confidences more accurately than GPT's own self-reported probabilities, suggesting biases in linguistic elicitation. These works highlight that correctness signals can transfer across models, but they largely remain in the one-model-to-one-model setting and do not study the factors that influence the calibration of a correctness model. By contrast, we document these factors and leverage the findings in our Generalized Correctness Model (GCM), which aggregates correctness patterns across many models, providing a more robust and empirically grounded calibration method.

**Downstream Applications.**    Correctness estimation has been leveraged to improve downstream tasks. Improved calibration benefits hallucination detection and truthfulness (Zhou et al., 2025; Li et al., 2024; 2025b), enhances interpretability (Stengel-Eskin et al., 2024a), strengthens reasoning ability (Wang et al., 2024b; Li et al., 2025a), improves semantic parsing (Stengel-Eskin & Van Durme, 2023), and supports reliable deployment in system-level routing setups (Hu et al., 2024; Wang et al., 2024a; Ong et al., 2024). Our GCM advances this line of work by providing a model-agnostic, history-aware framework for correctness estimation that generalizes across both models and datasets.

Table 25: **Conditioning factors** ablations (Section 3.3), full results on all metrics.

| Setting | Specific Model | | | | General Model | | | |
|---|---|---|---|---|---|---|---|---|
| | Acc | ECE | RMSCE | AUROC | Acc | ECE | RMSCE | AUROC |
| Full $P(c \mid q, r, \hat{r})$ | .7915 | .0171 | .0693 | .8570 | .8199 | .0231 | .0795 | .8904 |
| Answer-only $P(c \mid q, \hat{r})$ | .7451 | .0226 | .0867 | .8104 | .7892 | .0339 | .0880 | .8518 |
| Answerless $P(c \mid q)$ | .7043 | .0303 | .1006 | .7352 | .7197 | .0243 | .0948 | .7810 |

