# OpenReview forum: "Self-Knowledge Without a Self? Learning Calibrated and Model-Agnostic Correctness Predictors from Historical Patterns"
_ICLR.cc/2026/Conference — Submitted to ICLR 2026_

### Official Review · Reviewer_LRpD · 2025-10-29

**Soundness:** 4
**Presentation:** 3
**Contribution:** 3
**Rating:** 8
**Confidence:** 5

**Summary:**

The paper tests whether it is possible to fine-tune an LLM to predict correctness of itself or other LLMs, including multiple LLMs at once, and finds roughly equal performance in predicting an LLM’s performance by itself and other LLMs. They conduct several ablation experiments, including removing the answer and the response from the fine-tuning prompt, effectively fine-tuning the model to predict performance based on the query only, thus forcing it to extract patterns in the question space. Some experiments with in-context learning are also conducted.

**Strengths:**

- originality: the precise experimental setup (fine-tuning a model to predict correctness from various other models) is original, even though the broader idea has been around for a while
- quality: the experimental setup is sound and thorough, and so is the conceptualisation and interpretation of the findings
- clarity: the text is mostly clear
- significance: the fact that a model’s performance can be predicted as well by another model as by itself is interesting and insightful, particularly when the answer is not included; also, this finding seems to partly contradict that in https://arxiv.org/abs/2410.13787, which sets ground for interesting debate. Moreover, the strength of predictive power is significant. Also, evaluations on selective prediction is important.

**Weaknesses:**

- The paper could discuss more the real-world relevance of this for questions where the ground truth answer is unknown (and for which, therefore, the model fine-tuned to predict performance does not know it). What I mean is: performacne predictors that include the answer in the prompt may rely on knowledge of the ground truth, and the authors indicate how this likely explains the increased performance over not using the answer. However, this is only true when the answer is known to the model. There may be cases where the answer is unknown (to the model and the human users), and where performance prediction is therefore even more important. In practice, one can test the predictive methods on datasets where the performance of the model fine-tuned to predict performance of another one is very low (for instance, as the facts are after the knowledge cut-off of the model trained to predict performance, but not for the target model).
- I think an addressable weakness is not touching with a particular strand of work exploring the question of whether performance can be predicted, such as:
    - https://ojs.aaai.org/index.php/AAAI/article/view/21487 introduces the concept of “assessors”, small models trained to predict performance of a main one starting using the question only
    - https://ceur-ws.org/Vol-3169/paper4.pdf implements assessors for language models
    - https://arxiv.org/abs/2409.03563 develops assessors that work across multiple language models

    I invite the authors to indicate how these approaches relate to the one they propose in their related works section. I also believe that some of these approaches can be insightful baselines to compare their method with (for instance, using simple classifiers trained on top of sentence embeddings can further indicate if a LLM is needed as the predictive model, or if instead simpler approaches allow to determine predictive patterns in the prompt space).

- some minor clarity points:
    - lines 49-52 seem to suggest calibration is the only thing that matters for confidence, but this is not the case: one can have calibrated confidence by always predicting the success rate, with no discriminative power
    - it would be great to make more explicit the precise finetuning prompt(s) used in the main text, particularly how the answer, response, and model name are used (for instance, putting them in a figure, instead of hidden among the text)
        - for instance, the prompt in lines 167-168 asks how the model “will” respond to a prompt, but the authors say that this is appended to a prompt and model response. So why “will”? Is this what is actually used?

**Questions:**

- the authors’ findings seem to partly contradict those in https://arxiv.org/abs/2410.13787. How would the authors reconcile the two? Is there a substantial difference between the setups that may explain the difference in findings?
- lines 306-307: “These results indicate that correctness prediction generalizes across families, sizes, and even held-out stronger models.”. Is this due to models failing and succeeding in very similar ways?

---

> ### Author Response · Authors · 2025-11-21
> **Reviewer LRpD**
>
> Thank you for your comprehensive review and highlighting our findings as “**interesting and insightful**”, while also noting our “**sound and thorough**” conceptualization, experimental setup, and interpretation.
>
> **A1) discussion of scenarios when the answer is not known to the Correctness Model.** Thank you for raising this question \-- this scenario corresponds to an ablation of our method we refer to as the Answerless Correctness Model (introduced in RQ1 Setup). In this setting, we see potential that an Answerless CM could be used to understand whether to ask an LLM to generate a response to an answer (saving computation cost if it is expected that the LLM would be unable to answer), an Answerless CM trained on multiple models’ histories such as in Figure 3 (Answerless), might be able to evaluate which model will be most likely to answer a question correctly. **We have added a related works section to highlight this practical potential** in lines 502-510.
>
> **A2) relation to assessors.** Thank you for pointing out this connection to assessors. Our Answerless ablation setting is set up similarly to assessors, predicting correctness from just the query. **We have added a section on assessors to our related work in the updated pdf** spanning lines 502-510. *We’ll additionally follow up with an experiment using sentence embeddings rather than large LLMs to predict correctness.*
>
> **A3) minor clarity points. i)** We definitely agree that calibration is not the only thing that matters for confidence: this is why we refer to it as “a desirable quality for confidence” and measure accuracy and other metrics. We have made this more explicit in our introduction by highlighting that it is only one aspect (lines 51-53).  **ii)** We have clarified the phrasing of the prompts used to make it clear that “will respond correctly” is only used for the Answerless setting (lines 168-170). Although there are too many prompts to include in a reasonably sized figure, we have also added full prompts used for our different settings in Appendix D, and added a clear pointer in the main paper on line 239\.
>
> **Q1) paper with seemingly contradictory findings with RQ1.** Thanks for pointing out this interesting paper. A promising explanation is the difference between the type of self-knowledge we analyze. Their setup involves predicting the preference of models in preference prompts such as “should we charge customers for software updates” whereas ours involves predicting the correctness of a model when answering sometimes questions of varying difficulty. While models might have an distributional insight into what they are likely to output (e.g., being trained on text reflecting money seeking behavior may transfer to preferring the same behavior in a judge setting), LLMs are not trained on their historical correctness, which implies no privileged knowledge for our correctness prediction task.
>
> **Q2) models failing and succeeding in similar ways.** Yes, in RQ2b) we see generalization even in the Answerless setting and theorize that generalization occurs due to models failing or succeeding on similar questions (line 362), this could also be seen as learning some sort of intrinsic difficulty of a question. E.g., if the STEM split in MMLU usually contains more difficult questions for LLMs.
>
> **References:**
> \[1\] Pacchiardi, L. et al. PredictaBoard: Benchmarking LLM Score Predictability, 2025\. URL [https://arxiv.org/abs/2502.14445](https://arxiv.org/abs/2502.14445)
> \[2\] Zhou, L. et al. Reject Before You Run: Small Assessors Anticipate Big Language Models, 2022\. URL [https://link.springer.com/article/10.1007/s10994-024-06534-x](https://link.springer.com/article/10.1007/s10994-024-06534-x)

---

> ### Author Response · Authors · 2025-12-02
> **Followup Experiment for Reviewer LRpD**
>
> Dear reviewer LRpD,
>
> Following your question on the answerless ablation of our setting, we have conducted a new experiment in that setting to see if it is as effective to utilize a BERT model rather than an LLM in the answerless setting (bert-base-uncased). We observe that the efficacy of a BERT model is reduced compared to LLMs, but still usable, which is aligned with our conclusions in the paper. We hypothesize reduced performance using BERT could be due to the fact that the larger LLM still has more representational power, and differences between optimization (LoRA vs standard classifier head as you suggested). In the full setting, a bert model will perform much less well since predictive power scales with parametric knowledge, as shown in Appendix table 19, where we test effects of parametric knowledge.
>
> **Bert vs LLMs in the Answerless Ablation of Our Setting**
>
> | Configuration | Acc | ECE |
> | :---- | :---- | :---- |
> | Llama-3-8B \-\> Llama-3-8B | .694 | .016 |
> | BERT \-\> Llama-3-8B | .681 | .027 |

---

### Official Review · Reviewer_bn8A · 2025-11-01

**Soundness:** 1
**Presentation:** 4
**Contribution:** 2
**Rating:** 4
**Confidence:** 5

**Summary:**

This work investigates the problem of confidence calibration for Large Language Models (LLMs). The authors propose two research questions (RQs):

1. Are LLMs superior to other LLMs in predicting their own correctness?

2. What role does historical information from multiple models play in calibrated correctness prediction?

The study conducts experiments on open-source LLMs to address these questions.

**Strengths:**

1. The topic of LLM verbalized confidence calibration is highly important, and this research possesses significant practical relevance.

2. Both RQs are critical. A comprehensive solution to this issue would have a profound impact on the community, highlighting the paper's exceptional insight.

3. The authors designed a series of progressive experiments to substantiate their claims. The paper is well-structured and reads very clearly.

4. RQ2 is intuitive and is supported by convincing experimental evidence.

**Weaknesses:**

1. The validation for both RQ1 and RQ2 relies exclusively on logit-based confidence scores, which severely limits the paper's persuasive power. As noted in [1], logit-based confidence in large models can be unreliable due to the influence of RLHF. This is particularly problematic for RQ1: verbalized confidence generation is strongly dependent on a semantic understanding of the context. In contrast, logit confidence, which is based on probabilities adjusted by RLHF, is not semantically aligned with the concept of "confidence." Consequently, this approach fails to leverage the powerful In-Context Learning (ICL) capabilities of LLMs.

2. Following W1, the focus on logit-based scores restricts the study to smaller, weaker, open-source models. A critical claim like that in RQ1—which contradicts intuition and community consensus—requires validation on large-scale, state-of-the-art models (e.g., GPT-5, Claude 4, Gemini 2.5 Pro, DeepSeek V3.2) to be convincing. While the current experiments explore multiple angles (which is commendable), the most critical factors—model scale and quantity—are insufficient. The experiments for RQ1 are limited to Qwen2.5-7B and Llama3.1-8B. Even by small-model standards, this sample size is very small. A more appropriate study should include various sizes of Qwen, GPT-OSS models, Gemma, and others to provide a convincing basis for the conclusions.

3. The significance of "answerless" confidence is unclear. This metric seems to be a direct assessment of the question's intrinsic difficulty, which could be interpreted as a marginal distribution of confidence summed over all possible answers. However, this approach fails to account for the prior distributions resulting from the model's own predictions for specific answers. The rationale for using this as a basis for an ablation study needs more detailed justification.

[1] Katherine Tian, Eric Mitchell, Allan Zhou, Archit Sharma, Rafael Rafailov, Huaxiu Yao, Chelsea Finn, and Christopher Manning. 2023. Just Ask for Calibration: Strategies for Eliciting Calibrated Confidence Scores from Language Models Fine-Tuned with Human Feedback. In Proceedings of the 2023 Conference on Empirical Methods in Natural Language Processing, pages 5433–5442, Singapore. Association for Computational Linguistics.

**Questions:**

1. Lines 252-254: Why does removing the model response necessarily eliminate the influence of parametric knowledge?

2. Line 196: Is "Measuring Confidence" a typo here?

---

> ### Author Response · Authors · 2025-11-21
> **Response to Reviewer bn8A**
>
> Thank you for your detailed review of our paper and for noting that “**Both RQs are critical**”, “**highlighting the paper's exceptional insight**”.
>
> **A1) Usage of logit-based scores.** You point out an issue with our use of logit confidences for RQ1 and RQ2. As you have noted, logit confidences are indeed biased out of the box in RLHF-ed models. However, since we consider a training based regime and actively optimize the token probabilities, we are able to achieve negligible calibration error in our experiments for RQ1, as could be seen in Appendix Table 12 (\<2.5% ECE), which is superior to the calibration of verbalized methods (similar to \[1\]) benchmarked in our Table 2\. We also analyze whether models can directly take advantage of their ICL capabilities to improve the calibration of their predictions by referencing relevant performance history in context and show in Tables 2 and 3 that this is indeed the case for larger models (specifically Qwen3-32B). Please let us know if you meant something else in your comment regarding ICL capabilities.
>
> **A2) Additional models for RQ1.** Based on your feedback, we supplement our RQ1 results by repeating the experiment with the largest Llama3-70B and Qwen2.5-72B models. For both the full and answerless settings, we reach the same positive conclusion: that models are only as good at predicting other model’s confidences as they are predicting their own. Our analysis requires fine tuning, which makes the analysis of large closed source models difficult, but we plan to add models we can fine tune between the ranges of 3B to 72B to give more credence to this critical claim as you have suggested.
>
> | RQ1 Answerful Configuration                                | Acc   | ECE   |
> |----------------------------------------------|-------|-------|
> | Qwen2.5-72B → Qwen2.5-72B                    | .846  | .025  |
> | Meta-Llama-3-70B → Qwen2.5-72B               | .844  | .025  |
> | Qwen2.5-72B → Meta-Llama-3-70B               | .844  | .029  |
> | Meta-Llama-3-70B → Meta-Llama-3-70B          | .807  | .029  |
>
> | RQ1 Answerless Configuration                                | Acc   | ECE   |
> |----------------------------------------------|-------|-------|
> | Qwen2.5-72B → Qwen2.5-72B                    | .831  | .029  |
> | Meta-Llama-3-70B → Qwen2.5-72B               | .828  | .027  |
> | Qwen2.5-72B → Meta-Llama-3-70B               | .782  | .028  |
> | Meta-Llama-3-70B → Meta-Llama-3-70B          | .776  | .028  |
>
> **A3) Answerless setting.** Rather than only the intrinsic difficulty of the question, a Correctness Model (CM) trained in the answerless setting learns the general capabilities of a model, e.g., Qwen2.5-7B is not good at abstract algebra questions but consistently answers high school biology questions correctly. This setting allows us to equalize the playing field between stronger / weaker models and measure the presence of self-knowledge, without the interference of a larger model’s superior performance due to its greater knowledge of the true answers.
>
> **Q1) Removal of parametric knowledge.** By parametric knowledge, we mean when the Correctness Model knows information related to the true answer to a query, allowing it to use such knowledge to predict correctness of a given answer. By removing the answer from the response graded by the CM, the CM is only exposed to the question, negating the absolute advantage a larger CM may have over a smaller CM due to its knowledge of the true solutions. (Lines 255-262 in updated pdf)
>
> **Q2) Line 196 meaning.** This is a pointer back to the “measuring confidence” heading since it is hard to directly reference headings in LaTeX, we have put this in parentheses to make the intent clearer. (Line 198 in updated pdf)
>
> **References**:
> \[1\] Katherine Tian, Eric Mitchell, Allan Zhou, Archit Sharma, Rafael Rafailov, Huaxiu Yao, Chelsea Finn, and Christopher Manning. 2023\. Just Ask for Calibration: Strategies for Eliciting Calibrated Confidence Scores from Language Models Fine-Tuned with Human Feedback. In Proceedings of the 2023 Conference on Empirical Methods in Natural Language Processing, pages 5433–5442, Singapore. Association for Computational Linguistics.

---

> ### Author Response · Authors · 2025-11-27
> **Friendly reminder for reviewer bn8A**
>
> Dear reviewer bn8A,
>
> We wanted to briefly check in to see if you have had time to take a look at our reply, and if it has addressed your comments and will allow you to revisit your score. We have conducted critical **experiments on larger models** to justify RQ1 by finetuning Qwen2.5-72B, Llama-3-70B and Gemma-3-27B, and replied to your other questions. This followup additionally includes new results involving the Gemma-3 series of models.
>
> To make our tests on larger and varied models more comprehensive, we conducted new nxn tests between Qwen2.5-72B, Llama-3-70B, and Gemma-3-27B (the largest variants of each open source model), showing that there is no particular advantage for self-prediction (accuracy differences \<1% in the answerless setting), aligning with the findings in our paper.
>
> **Additional Large Models Experiments**
> **Accuracy**
> | Target Model \\ Correctness Model | Qwen-2.5-72B   | Llama-3-70B  | Gemma-3-27B |
> | :------------------------------- | :----- | :----- | :---- |
> | Qwen-2.5-72B                             | .831   | .828   | .827     |
> | Llama-3-70B                           | .782   | .776   | .774     |
> | Gemma-3-27B                            | .784   | .784   | .791   |
>
> **ECE**
> | Target Model \\ Correctness Model | Qwen-2.5-72B   | Llama-3-70B  | Gemma-3-27B |
> | :------------------------------- | :----- | :----- | :---- |
> | Qwen-2.5-72B                               | .029   | .027   | .031     |
> | Llama-3-70B                            | .028   | .028   | .037     |
> | Gemma-3-27B                             | .030   | .026   | .029    |
>
> Additionally, to explore further impacts of model scale as you suggested, we conducted tests in the full setting between 4 sizes of Gemma, matching the conclusions in our paper (no notable advantage for self-prediction and models with more parametric knowledge outperforms smaller models).
>
> **Additional Gemma Size Experiments**
> **Accuracy**
> | Target Model \\ Correctness Model | Gemma-3-4B | Gemma-3-12B | Gemma-3-27B |
> | :------------------------------- | :--------- | :---------- | :---------- |
> | Gemma-3-4B                       | .687       | .817        | .831        |
> | Gemma-3-12B                      | .772       | .796        | .829        |
> | Gemma-3-27B                      | .794       | .812        | .825        |
>
> **ECE**
> | Target Model \\ Correctness Model | Gemma-3-4B | Gemma-3-12B | Gemma-3-27B |
> | :------------------------------- | :--------- | :---------- | :---------- |
> | Gemma-3-4B                       | .028       | .031        | .017        |
> | Gemma-3-12B                      | .026       | .028        | .022        |
> | Gemma-3-27B                      | .025       | .025        | .022        |
>
> With the addition of these tests, our RQ1 results now involve Gemma-3-27B, Gemma-3-12B, Gemma-3-4B, Mistral-7B, Qwen3-8B, Llama3.1-8B, Qwen2.5-7B, Qwen2.5-72B, and Llama-3-70B (5 model families and various sizes). The full array of tests have been added in *Appendix Tables 9 to 22*. Please let us know if our responses have addressed your comments, and if you have any more questions or suggestions.

---

### Official Review · Reviewer_WbFq · 2025-11-03

**Soundness:** 2
**Presentation:** 3
**Contribution:** 3
**Rating:** 2
**Confidence:** 3

**Summary:**

This paper argues that a model 1) _has no privileged information about its own error distribution_ and hence that it is more effective to 2) _treat the ability to predict correctness / detect error as a downstream task on its own_. (Here I am wording the two points my own way,  hopefully the authors will agree that my phrasing represents their views well.)

The paper provides evidence for (1) by testing whether a model can predict its own correctness any better than some other model could, the observation is in the negative. The paper goes over a number of design ideas to essentially deliver (2), for which of course it uses data about a model's error distribution (that is, annotation for when a model has been wrong/correct in the past). The paper shows that by engineering this component (the paper calls it a 'correctness model') carefully, the ability to predict correctness can even generalise across a number of conditions.

The paper is generally well-written, the research is interesting and the findings are likely to be echoed / built upon. That said, I do find it to miss an entire branch of relevant literature, but I will comment on that in weaknesses.

**Strengths:**

1. RQ1, both its formulation and the simplicity of the observation that supports it is very refreshing.

I personally think it's about time that someone dispels the tacit assumption that models have privileged information outside the merits of carefully engineered heuristics or uncertainty quantifiers (the 'careful' in it being the designer's evolving knowledge - paper after paper - of what works and what doesn't, effectively, a proxy to learning from historical data). For that, I find this paper rather refreshing.

2. I have a problem with how the paper positioned RQ2 (see weakness), but the strategies developed to approach it (the various designs for the so-called 'generalised correction models' and the observations in the various settings) come across as quite thorough.

**Weaknesses:**

My main problem with this paper is that it is written as if confidence and quality estimation had never been approached as ML tasks ever before. Learning from 'historical data about correctness' is precisely what these fields have always done (from SVMs, to Bayesian models, to neural models, to LLMs, with supervision from a single model or via system combination, with and without feature engineering, etc; confidence and quality estimation have been approached by everything in the ML toolbox). The oldest reference I can think of on the spot is [Specia's 2009 paper](https://aclanthology.org/2009.eamt-smart.10/), but please do the literature check, you will uncover more papers than I can list  (from small university labs to the biggest industry labs out there) and you will find that many of these papers are hugely impactful.

To the best of my judgment, the research in this submission stands, but this paper - even if inadvertently - currently obfuscates that  "learning from historical data about correctness" is an actual thing and it's been called QE for at least nearly two decades.

I am not sufficiently close to QE myself, so I cannot tell you whether the designs proposed in the paper as 'correctness models' are at all surprising. I do suspect they aren't, but I will refrain from penalising in that dimension. Instead I will just hope that there's a QE reviewer in the loop. I am however penalising for a kind of obfuscation of literature that I consider harmful.

**Questions:**

My expectation is that the paper should be transparent / clear rather upfront about the point above, namely, the framing of confidence/quality estimation as a task on its own, for which models can be trained on historical data, etc, is generally accepted, well-established and _not_ a contribution of this paper.

Of course, in light of the findings surrounding RQ1, the importance of embracing that framing is clear, and this is an argument you are establishing and supporting in the submission. I appreciate that and am not challenging it.

Last, in your literature check, I'd recommend looking for recent papers (for example from Unbabel or WMT submissions to the QE task; these are just some entry points into modern QE) as the concrete designs you propose might have been proposed before and, I hope you agree, it is only right to give credit where credit is due (even if, incidentally, those papers did not inspire you directly).

---

> ### Author Response · Authors · 2025-11-21
> **Response to Reviewer WbFq**
>
> Thank you for your review and for highlighting our “**well-written**” and “**interesting**” research that is “**likely to be echoed / built upon**”.
>
> **A1) Connection to Quality Estimation.** While training an estimator to predict the quality of outputs is similar to our setting, the key difference is that Quality Estimation (using Machine Translation (MT) as an example) frames the task as predicting machine translation quality given only the original text and a potential translation, explicitly without reference to what model generated the text, in order to serve as an unbiased evaluator \[1,2,3,4\]. More recently, similar approaches have been attempted on many tasks in the form of "LLM-as-a-Judge" frameworks also without information about the generating model \[5, 6\].
>
> In contrast, we train on “historical information from multiple models”, this means conditioning on the identity of the models generating a response, as well as training on an unaltered history of model performance. Concretely, this allows us to learn both tasks that a specific model is strong/weak in as well as the general difficulty of types of questions for LLMs (e.g., we find in RQ2b that a correlation between errors of different models gives rise to further generalization).
>
> Practically, in questions where a QE/Judge cannot verify the correctness/quality of a response directly (e.g., verifying the correctness of responses to an extremely difficult programming question), a Correctness Model (CM) will still be able to achieve nontrivial performance by recalling the historical performance of the target model on similar tasks (for example, an answerless CM operating solely based on historical performance achieves \+22% accuracy above random, Figure 3).
>
> Further, quality estimation approaches generally do not produce or analyze a calibrated confidence estimate in our sense (in line 50, we define confidence to mean predicting 90% confidence on an output should correspond to a 90% chance of the output being correct). This is because the ground truth for quality estimation usually does not have probabilistic meaning in the way we consider (e.g., MT where direct assessment uses a human labeled quality of translation from 1-100 points \[3\]). Related works such as reward models and LLM-as-a-Judge frameworks also generally do not address “calibrated” confidences in our sense \[5, 6\].
>
> **We thank you for pointing out this connection, and have added a new related works section spanning lines 481-500 to address connections to Quality Estimation and related approaches.**
>
> **References**:
> \[1\] Specia, L. et al. Sentence-level confidence estimation for MT, May 2009\. URL [https://aclanthology.org/2009.eamt-smart.10.pdf](https://aclanthology.org/2009.eamt-smart.10.pdf)
> \[2\] Zhao, H. et al. From Handcrafted Features to LLMs: A Brief Survey for Machine Translation Quality Estimation, March 2024\. URL [https://arxiv.org/pdf/2403.14118](https://arxiv.org/pdf/2403.14118)
> \[3\] Zerva, C. et al. Findings of the Quality Estimation Shared Task at WMT 2024 (Ninth Conference on Machine Translation), November 2024\. URL [https://aclanthology.org/2024.wmt-1.3.pdf](https://aclanthology.org/2024.wmt-1.3.pdf)
> \[4\] Moosa, I. M. et al. MT-Ranker: Reference-free Machine Translation Evaluation by Inter-System Ranking, January 2024\. URL [https://arxiv.org/pdf/2401.17099](https://arxiv.org/pdf/2401.17099)
> \[5\] Tan, S. et al. JudgeBench: A Benchmark for Evaluating LLM-Based Judges, October 2024\. URL [http://arxiv.org/abs/2410.12784](http://arxiv.org/abs/2410.12784)
> \[6\] Zheng, C. et al. A Survey of Process Reward Models: From Outcome Signals to Process Supervisions for Large Language Models, 2025\. URL [https://arxiv.org/abs/2510.08049](https://arxiv.org/abs/2510.08049)

---

> ### Author Response · Authors · 2025-11-26
> **Friendly reminder for reviewer WbFq**
>
> Dear reviewer WbFq,
>
> We wanted to briefly check in to see if you have had time to take a look at our reply, and if it has addressed your comments and will allow you to revisit your score. We have **added a related work section** and discussed in our response the **connections and differences between our work and Quality Estimation**. If any part of our response would benefit from further detail before the discussion period ends, we would be very happy to elaborate.

---

### Official Review · Reviewer_CcM8 · 2025-11-09

**Soundness:** 2
**Presentation:** 2
**Contribution:** 1
**Rating:** 2
**Confidence:** 4

**Summary:**

The authors analyzed the judgment accuracy of response content by both their own models and other models, and found that models do not exhibit higher accuracy in judging their own answers—instead, more capable models demonstrate higher accuracy in judging answers. Meanwhile, the authors trained a judgment model and observed that it could achieve higher judgment accuracy. They argue that this finding indicates model-related issues can be optimized by leveraging certain patterns and confidence levels to assess the response accuracy of models. Additionally, the authors suggest that both post-hoc calibration and in-context learning (ICL) are effective in improving accuracy.

**Strengths:**

The authors' writing is highly concise and accessible.

The topic of confidence is of great importance, especially in the context where hallucinations are becoming increasingly severe under the reinforcement learning paradigm brought by o1/R1.

**Weaknesses:**

Although the authors focus on the confidence field, in the reviewer's opinion, they fail to conduct sufficient literature research.

On one hand, the authors overlook the field of reward models. The Correctness Model proposed by the authors appears to be a type of reward model, yet there are numerous open-source models in this domain—including General Reward Models and Process Reward Models. Notably, many existing studies in the reward model field have already reached similar conclusions to those presented in this work. But there is no baseline from reward models.

Despite the rigorous logical flow of the research content, its innovation and inspirational value are, in the reviewer's view, insufficient to support its acceptance by top-tier conferences.

**Questions:**

As stated in Weakness

---

> ### Author Response · Authors · 2025-11-21
> **Response to Reviewer CcM8**
>
> Thank you for your review and for highlighting the “**great importance**” of our focus on confidence estimation as well as our “**highly concise and accessible**” writing.
>
> **A1) connection to reward models.** While we suggest in our introduction (line 47-48) that calibrated confidences could be leveraged as a RL signal similar to rewards from reward models, we operate in a different setting. *By design, reward models do not consider historical information as they are trained with no knowledge of which model produced the answers they are judging \[1,2\].* This is necessary to produce unbiased reward models. In contrast, we focus on learning from “historical patterns”. Additionally, the work on reward models generally does not consider the confidence calibration of the model’s judgements \[3,4,5\].
>
> To clarify the difference in settings, consider any case where the CM/RM/Judge has almost no parametric knowledge about the true answer (e.g., given an extremely difficult math problem, or an extremely obscure trivia question), the RM performance would reduce to random as parametric understanding approaches zero \[2\], whereas the Answerless GCM in Figure 3 shows the nontrivial performance of a CM (+22% above random). With historical patterns, it is possible to learn both tasks that a specific model is strong/weak in as well as the general difficulty of various tasks.
>
> Additionally, our focus on analyzing the confidence calibration of CMs is in contrast to LLM reward models that do not analyze calibrated confidences. Reward models generally produce unbounded scalar rewards that do not correspond to statements about probability of correctness \[1, 4\], Judge models commonly produce likert scale judgements \[2, 5\], for verifiers and reward models that can output probabilities such as \[3, 4\], calibration is generally not considered.
>
> Thank you for pointing out this connection, **we have added a new heading in the related works section, spanning lines 481-500, to discuss our connection with reward models**. We briefly summarize the discussed qualitative differences between settings in the table below.
>
> **Differences between what Reward Models and Correctness Models Condition On**
>
> | Model Type | Query | Response | Identity of Target Models | Unaltered Performance History |
> | :---- | :---- | :---- | :---- | :---- |
> | Reward Model (RM) | Y | Y | N | N |
> | Correctness Model (CM) | Y | Y | Y | Y |
>
> **References:**
> \[1\] Lambert, N. et al. RewardBench: Evaluating Reward Models for Language Modeling, March 2024\. URL [http://arxiv.org/abs/2403.13787](http://arxiv.org/abs/2403.13787)
> \[2\] Tan, S. et al. JudgeBench: A Benchmark for Evaluating LLM-Based Judges, October 2024\. URL [http://arxiv.org/abs/2410.12784](http://arxiv.org/abs/2410.12784)
> \[3\] Luo, L. Improve Mathematical Reasoning in Language Models by Automated Process Supervision, June 2024\. URL [http://arxiv.org/abs/2406.06592](http://arxiv.org/abs/2406.06592)
> \[4\] Wang, Z. et al. RLBFF: Binary Flexible Feedback to Bridge Between Human Feedback & Verifiable Rewards, September 2025\. URL [http://arxiv.org/abs/2509.21319](http://arxiv.org/abs/2509.21319)
> \[5\] Kim, S. et al. Prometheus: Inducing Fine-Grained Evaluation Capability in Language Models, 2023\. URL [http://arxiv.org/abs/2310.08491](http://arxiv.org/abs/2310.08491)

---

> ### Author Response · Authors · 2025-11-26
> **Friendly reminder for reviewer CcM8**
>
> Dear reviewer CcM8,
>
> We just wanted to briefly check in to see if you have had time to take a look at our reply, and if it has addressed your comments and will allow you to revisit your score. We have **added a related work section** and talked about the **similarities and differences between our setting and the setting for reward models**. If any part of our response would benefit from further detail before the discussion period ends, we would be very happy to elaborate.

---

### Author Response · Authors · 2025-12-02
**Summary for Area Chair**

This note summarizes the reviewer consensus, questions, and the concrete improvements we made during the rebuttal period. We responded to all reviewer questions, strengthened empirical validation of RQ1 with substantially larger and more diverse models, and clarified positioning by adding and sharpening our related-works discussion on reward models and quality estimation.

## **Reviewer Consensus**

* **Novelty and Insight:** Our findings are highlighted as “**interesting and insightful**” (LRpD), **updates** “**intuition and community consensus”** (bn8A), and “**dispels the tacit assumption**” of privileged information for confidence estimation (WbFq). Reviewer (bn8A) also writes “**Both RQs are critical**”, “**highlighting the paper's exceptional insight**”.
* **Importance and Impact:** Reviewer (CcM8) highlights the great importance/timeliness of our topic of confidence estimation and our “**highly concise and accessible**” writing.  Reviewer (WbFq) writes that they are “**likely to be echoed / built upon**”. Reviewer (bn8A) highlights our topic as “**highly important**” with “**significant practical relevance**”, and reviewer (LRpD) notes that our inclusion of downstream “**evaluations on selective prediction is important**.”
* **Clarity:** Reviewers highlight that “**The paper is well-structured and reads very clearly**” (bn8A) and is “**highly concise and accessible**” (CcM8).

## **Rebuttal Improvements**

### **Additions for Reviewer bn8A**

**Additional Experiments**

* W2: Asked for further validation on RQ1 with larger and more varied models beyond the 7B scale.
* A2: Results from ***7 new experiments*** **finetuning larger and more varied LLMs aligned with the conclusions in our paper** were added as Tables 15 to 22 in the updated pdf. We want to highlight our efforts in finetuning the Llama3-70B, Qwen2.5-72B, and Gemma-3-27B models (the largest models in their families), and further experiments with all sizes of Gemma-3.

**Clarification Questions**

* W1: Worried that our focus on raw logit confidences may have higher calibration error compared to verbalized confidences in RQ1.
* A1: Clarified that RQ1 uses a training based setting and we showed this overcomes the issue, achieving lower calibration error than verbalized confidences. We additionally highlight our extensive exploration of verbalized confidences in RQ2.
* W3: Answerless setting seems only to assess intrinsic question difficulty.
* A3: We clarified that the answerless setting allows learning of model specific performance characteristics, such as the performance of Qwen-3-8B on math vs biology questions.
* We additionally responded to all their smaller clarity questions, and we invite your review of our responses marked “Q1” and “Q2” in our reply.

### **Additions for Reviewer CcM8**

**Related Works**

* W1: Request literature review of our work’s similarity to reward models, interpreting our Correctness Models (CMs) as reward models.
* A1: We expanded our related works section to highlight the connections and differences between reward models and CMs (lines 481-500). **We differentiated the settings we operate under, describing clear cases where CMs will outperform reward models** (**\+22%** in case of low parametric knowledge). Additionally, we highlighted our focus on historical patterns and probabilistic confidence calibration in contrast to work in reward models.
* No other questions were presented.

### **Additions for Reviewer WbFq**

**Related Works**

* W1: Asked for a related works section on Quality Estimation, which like reward models can be framed as llm-as-a-judge.
* A1: **We highlighted the specific similarities and differences and included quality estimation in our expanded related works section as requested (lines 481-500)**, and we showed the difference in our settings and when CMs would outperform quality estimation.
* No other questions were presented.

### **Additions for Reviewer LRpD**

**Related Works**

* W1,2: Reviewer LRpD was excited about the “Answerless” ablation of our setting analyzed in our paper and encouraged us to highlight the practical consequences of this ablation. They also mentioned potential connection of the Answerless ablation to Accessor models.
* A1,2: We highlighted Accessor models and practical applications of the Answerless ablation our expanded related works (Lines 502-510).

**Additional Experiment**

* W2: For the answerless ablation, reviewer LRpD was curious if a BERT model can also be used instead of an LLM, since the answerless setting did not involve parametric knowledge.
* A2: We conducted an experiment showing the performance of the BERT model when used as the predictor for the answerless ablation setting, showing slightly lowered but nontrivial performance, which lines up with our expectations.
* We also addressed all of their questions, please check Q1, Q2 in our response. We also addressed minor writing clarity points in A3.

---

### Meta-Review · Area_Chair_Eezk · 2026-01-07

**Summary:**

This paper analyses the question about whether a language model has privileged information about its own error distribution and thus is better at predicting its own confidence rather than having an external model predict confidence.  In the analysis the authors conclude that no, a model doesn't predict its own correctness any better than another model trained on historical data from this model.  The authors assess a model's confidence through using p(true) on the model prompted to answer 'yes' or 'no' about the correctness of a response.

The reviews are somewhat polarized, with one champion giving an 8, two rejects (2) and a borderline reject (4).  The reviewers in general found the paper well written and interesting.  They seemed to find the research question compelling and the findings potentially useful and interesting.
The two most negative reviews (reviewers CcM8 and WbFq) seemed to be primarily focused on novelty, with one having concerns on connections to reward modeling and another on extensive literature on confidence estimation.

Reviewer bn8A seemed most concerned that the authors used only logit-based confidence scores (a concern that I agree with). There is now a lot of literature on confidence estimation for LLMs and logit-based P(True) is not necessarily SOTA.  I'd be very curious to understand if measures of semantic entropy convey similar results.  In any case, it seems risky to make a definitive negative conclusion about an LLMs ability to self-assess its own confidence without ablating over different strategies for eliciting the confidence measure.   Nevertheless, this does seem like an interesting result.

I appreciate the detailed responses that the authors gave to the reviewers' concerns, and the additional experiments and results they obtained.  Unfortunately, the paper was quite significantly below the bar in terms of average review scores, weighted down by two 2's.  Unfortunately, it seems like the revisions would have to be fairly extensive, beyond adding more discussion of related work and some more experiments, to sway the reviewers to change their scores enough to get to a majority accept or accept average.  However, the reviewers are clearly signaling that they see a lot of value in this work and that there are many positive aspects and takeaways.  Rather it seems that the paper is just not quite ready for publication.  Hopefully the reviews and the additional effort put in by the authors will strengthen the paper for a future submission.

**Reviewer Concerns:**

Reviewers CcM8 and WbFq were concerned about novelty compared to the existing literature. The authors added more literature review and discussion of existing work.  This goes some way to addressing the reviewers' concerns.  It seems like the reviewers were asking for signficant re-framing of the contribution of the paper (over-claiming) and still had concerns that the work was similar to existing results.  I think the reviewers may likely have raised their scores by a point or so, but I don't believe it would be sufficient to push to an accept score (a jump of at least 3-4 points).

Concerns about experimental results have been allayed somewhat with a significant number of additional experimental results.  This might sway bn8A to raise their score by one.

The authors argue that their logit-based confidence scores are well-calibrated and better empirically than their verbalized confidence estimates.  I'm not sure this fully addresses this concern, given that there remain a variety of methods for assessing model confidence.

**Reviewer Scores:**

I think CcM8, WbFq and bn8A would plausibly each have raised their score by 1.

---

### Decision · Program_Chairs · 2026-01-26

Reject